# Decentralized Data Collection for Robotic Fleet Learning: A Game-Theoretic Approach

**Oguzhan Akcin**[1], **Po-han Li**[1], **Shubhankar Agarwal**[1] **and Sandeep P. Chinchali**[1] *

**Abstract:** Fleets of networked autonomous vehicles (AVs) collect terabytes of sensory data, which is often transmitted to central servers (the "cloud") for training machine learning (ML) models. Ideally, these fleets should upload all their data, especially from rare operating contexts, in order to train robust ML models. However, this is infeasible due to prohibitive network bandwidth and data labeling costs. Instead, we propose a cooperative data sampling strategy where geo-distributed AVs collaborate to collect a diverse ML training dataset in the cloud. Since the AVs have a shared objective but minimal information about each other's local data distribution and perception model, we can naturally cast cooperative data collection as an $N$-player mathematical game. We show that our cooperative sampling strategy uses minimal information to converge to a centralized oracle policy with complete information about all AVs. Moreover, we theoretically characterize the performance benefits of our game-theoretic strategy compared to greedy sampling. Finally, we experimentally demonstrate that our method outperforms standard benchmarks by up to 21.9% on 4 perception datasets, including for autonomous driving in adverse weather conditions. Crucially, our experimental results on real-world datasets closely align with our theoretical guarantees.

## 1 Introduction

Envision a fleet of autonomous vehicles (AVs) that observes heterogeneous street scenery, weather conditions, and rural/urban traffic patterns. To train robust ML models for perception or trajectory prediction, these AVs should share as much diverse fleet data as possible in the cloud, while balancing network bandwidth, data storage, and labeling costs.[2] Given these constraints, we argue that AVs must *coordinate* how to sample rare, out-of-distribution (OoD) data with common examples based on their unique local data distributions. For example, if only a few AVs operate in heavy snow, they should specialize in sending snowy images to the cloud, while others should send data from more common scenarios like sunny weather. Since the AVs have a shared target data distribution (objective) but limited information on each other's local data distribution and potentially private ML models, our key contribution is to cast data collection as a **N-Player mathematical game**.

In our game-theoretic formulation (Fig. 1), the AVs exchange minimal information to choose a data sampling strategy (what limited subset of data-points to upload). Importantly, we prove that an AV fleet will quickly converge to a **Nash equilibrium** (i.e., a fixed point where each robot does not change its sampling strategy) [3, 4] with bounded communication. Morever, our practical formulation accounts for perceptual uncertainty from *imperfect* computer vision models and heterogenous local data distributions. As such, to the best of our knowledge, we are the first to cast data sampling from networked robots as a mathematical game. In summary, our key contributions are:

1. We provide a novel formulation for distributed data collection as a *potential* game [5] since the robots attempt to minimize a common convex objective function that incentivizes them to reach a balanced target data distribution in the cloud. We prove that our strategy converges to a centralized oracle policy and, under mild assumptions, converges in a single iteration.

*[1] Department of Electrical and Computer Engineering (ECE), The University of Texas at Austin, Austin, TX {oguzhanakcin,pohanli}@utexas.edu, {somi.agarwal,sandeepc}@utexas.edu

[2] A single AV can measure over 20-30 Gigabytes (GB) per second of video and LiDAR data [1] while a typical 5G wireless network only provides 10 Gbps of bandwidth for *multiple* users [2].

6th Conference on Robot Learning (CoRL 2022), Auckland, New Zealand.

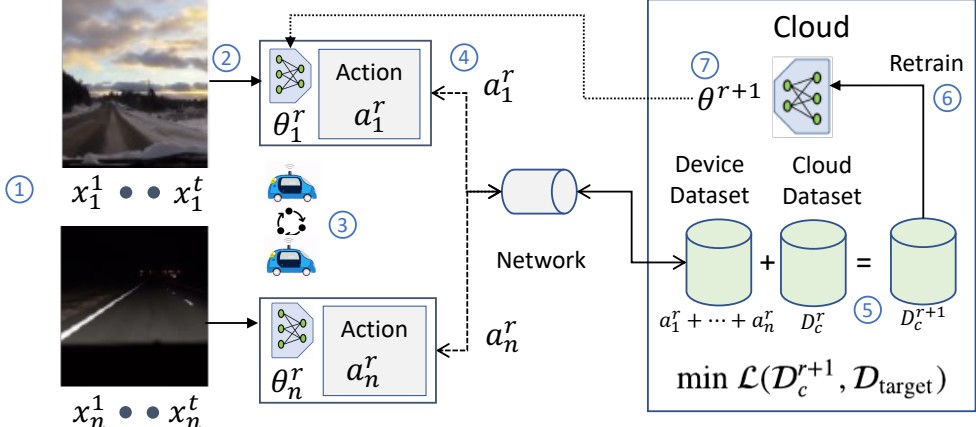

Figure 1: **Game-Theoretic Data Collection:** Each step in our cooperative algorithm is numbered in blue. First, each AV $i$ observes a sequence of images $x_i^t$ in each round $r$ of data collection (step 1). Then, it classifies each image $x_i^t$ with a local vision model with parameters $\theta_i^r$ (step 2). Then, it samples a limited set of $N_{\text{cache}}$ images according to its action policy $a_i^r$, which governs what distribution of data-points to upload. Crucially, the action $a_i^r$ is chosen cooperatively with other AVs using a distributed optimization problem (step 3). Next, each AV transmits its local cache of data-points to the cloud (step 4). The current cloud dataset, $\mathcal{D}_c^r$, is updated with the new uploaded data-points $a_i^r$ (step 5). The combined cloud dataset, $\mathcal{D}_c^{r+1}$, can be used to periodically re-train new model parameters $\theta^{r+1}$ (step 6), which are then downloaded by the AVs (step 7). All AVs share a goal of minimizing the distance between the collected cloud dataset $\mathcal{D}_c^{r+1}$ (green) and the target $\mathcal{D}_{\text{target}}$.

2. We provide theoretical performance bounds characterizing the benefits of our game-theoretic approach compared to greedy, individual behavior.
3. We show that our proposed strategy outperforms competing benchmarks by 21.9% on 4 datasets, including the challenging Berkeley DeepDrive autonomous driving dataset [6].

**Related Work:**   Data collection from networked robots is related to cloud robotics [7–15] and active learning [16–20]. In such prior works, robots either send *all* their data to the cloud or select samples individually without coordination. In contrast, we exploit the fact that networked AVs can coordinate how to sample rare data to achieve a better outcome (i.e., balanced data distribution).

Federated learning (FL) [21–29] enables a fleet of mobile devices to train ML models on local private data and only share anonymized gradient updates with the cloud. However, our work is fundamentally different, and even complementary, to standard FL. First, FL makes the restrictive assumption that each robot has perfectly labeled local data, which is infeasible for AVs that observe rare, OoD images. Instead, we address a practical scenario where robots run local inference with only an *imperfect* vision model that guides data collection. Moreover, FL does not statistically sample data but trains on all of it locally, while our approach only uploads a limited set of images to reduce network and data labeling costs. Finally, we assume robots only receive ground-truth labels for the uploaded data in the cloud, which is required for training on rare classes.

Our setting is a non-cooperative game since the robots do not explicitly form coalitions and act with minimal information about each other [5, 30–34]. Specifically, our setting is a potential game since each robot attempts to maximize a shared concave objective function (the common *potential* function) that rewards progress towards a balanced target data distribution in the cloud. As detailed in Sec. 2 and Appendix 5.1, changes in the common potential function directly translate to changes in each robot's policy towards a Nash Equilibrium. While concave games have been applied to problems such as wireless network resource allocation [35], ours is the first work to contribute a game-theoretic formulation for distributed data collection from a fleet of robots.

## 2   Problem Formulation

We now formulate a practical scenario, shown in Fig. 1, where distributed robots collect data to train a robust ML model in the cloud. Our goal is to select an appropriate *action* for each robot, specifically the data-points it should upload, so that the overall collected cloud dataset closely matches a given target, such as an equal distribution over all classes. Fig. 2 intuitively depicts data sampling.

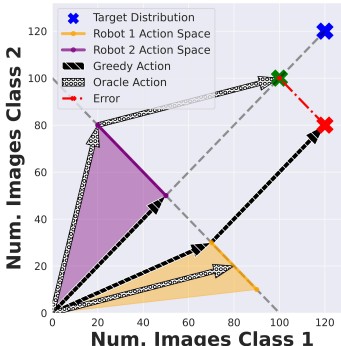

Figure 2: **Why Cooperate?** Consider a toy example with only 2 classes and 2 robots. The axes represent the number of data-points for each class. Our goal is to reach the target distribution (blue cross) where each class has 120 data-points, represented by $(120, 120)$. The robots start at $(0,0)$ with no data-points in the cloud. The possible combinations that can be uploaded from robots 1 and 2 are shown as the shaded feasible action spaces (yellow and purple). This shaded region is determined by the robot's local data distribution and vision model accuracy (Def. 2). GREEDY (black) individually calculates the projection of the target distribution onto each robot's feasible action space, but the sum of actions may not be optimal, leading to a high error (red). However, OR-ACLE accounts for the two robots' action spaces and thus minimizes the error between the target dataset and the sum of actions (grey).

Our general formulation applies to any robot constrained by network, storage, or labeling costs, ranging from Mars Rovers constrained by the Deep Space Network ($< 5$ Mbps) [36] or AV fleets.

**Robot Perception Model:** For a simple exposition, we first consider a general computer vision classification task with $N_{\text{class}}$ classes. The dataset used for training the model is stored in the cloud. Each period of data collection, such as a day, is denoted by a *round r* and data is uploaded to the cloud at the end of a round $r$. The cumulative dataset stored in the cloud at the end of round $r$ is denoted by $\mathscr{D}_c^r$, whose size is given by $N_{\mathscr{D}_c^r} = |\mathscr{D}_c^r|$. $N_{y_j}$ denotes the number of class $j$ data-points in the dataset $\mathscr{D}_c^r$. Therefore, the distribution of classes in the dataset $\mathscr{D}_c^r$ is denoted by $\rho_{\mathscr{D}_c^r} = \left[ N_{y_0}, N_{y_1}, \cdots, N_{y_{N_{\text{class}}}} \right]$. Each robot $i$ has a perception model, such as a deep neural network (DNN), where local inference is denoted by $\hat{y} = f(x; \theta_i^r)$. Here, $f(\cdot; \theta_i^r)$ is a model with parameters $\theta_i^r$ at round $r$, $\hat{y}$ is the predicted label for input $x$, and $y$ is the corresponding ground-truth label.

Importantly, the models can be *imperfect* – each model has a confusion matrix, $C_i^r \in \mathbb{R}^{N_{\text{class}} \times N_{\text{class}}}$ (Eq. 4) that captures the probability of predicting class $\hat{y}_j$ for an image with true class $y_j$, denoted by $p(\hat{y}_j | y_j)$. In practice, one of the $N_{\text{class}}$ classes can represent an "unknown" category while the rest of the $N_{\text{class}} - 1$ classes can represent a mixture of rare and well-understood concepts. Further details on the confusion matrix are provided in Appendix 5.2. Finally, while we use a (likely imperfect) classification model to sample images, the uploaded data can be used to train models for diverse tasks such as object detection, semantic segmentation etc.

**Robot Fleet:** We consider a fleet of $N_{\text{robot}}$ robots, where each robot $i$ collects a data-point $x_i^t$ at time $t$ from its local environment (i.e., camera image or LiDAR scan). The distribution of true classes observed by a robot $i$ in round $r$ is denoted by $p_i^r(y) \in \mathbb{R}_+^{N_{\text{class}}}$, which sums to one over the $N_{\text{class}}$ classes. From this distribution, a robot $i$ observes a large dataset of images on round $r$ denoted by $\mathscr{D}_i^r$ of size $|\mathscr{D}_i^r| = N_i^r$. However, to limit network bandwidth and data labeling costs, each robot $i$ can only upload $N_{\text{cache}} \ll N_i^r$ images to the cloud at the end of round $r$, which it stores in an on-board cache within the round. The size of $N_{\text{cache}}$ can be flexibly set by a roboticist based on data upload and labeling budgets. The class predictions, $\hat{y}_j$, are generated by running local inference of the classification model, $\hat{y}_i = f(x_i^t; \theta_i^r)$, for the collected data-points $x_i^t$. Finally, $p_i^r(\hat{y})$ denotes the distribution of *predicted* classes observed by robot $i$.

**Assumption 1.** *The number of data-points collected by a robot on any round $r$, $N_i^r$, is significantly greater than the size of the local robot cache, $N_i^r \gg N_{cache}$.*

This is a valid assumption since each robot will collect much more data compared to the amount it can economically upload. Our formulation is extremely general – each robot can have different (or the same) model parameters $\theta_i^r$ and observe a different distribution $p_i^r(y)$ of the $N_{\text{class}}$ classes.

**Robot Statistical Sampling Action:** At each round $r$, each robot $i$ takes an action which determines how many data-points of each class to send to the cloud. We define each robot $i$'s action at round $r$ as $a_i^r = \left[ N_{y_0}, N_{y_1}, \cdots, N_{y_{N_{\text{class}}}} \right]$, i.e. the number of data-points of each class $j$. Our key technical innovation will be to illustrate *how* to cooperatively select an optimal action. Importantly, since each robot $i$ has an *imperfect* perception model with confusion matrix $C_i^r$, there is uncertainty in the effect of taking any action $a_i^r$. As such, our natural next step is to define the set of feasible actions any robot can upload given its local data distribution and perceptual uncertainty.

**Definition 1** (Feasible data matrices). *A feasible data matrix, $P_i^r \in \mathbb{R}^{N_{class} \times N_{class}}$, of robot i in round r is the probability matrix defined as:*

$$P_i^r = [p_{i,1}^r, ..., p_{i,N_{class}}^r],$$

*where $p_{i,j}^r = \frac{C_{i,j}^r * p_i^r(y)}{\|C_{i,j}^r * p_i^r(y)\|_1} = p(y|\hat{y}_j) \in \mathbb{R}^{N_{class}}, \forall j = 1, ..., N_{class}$. We use $*$ as element-wise multiplication of vectors, $\|\cdot\|_1$ as the $L_1$ norm, and the second subscript j to denote the j-th column of a matrix. We assume $P_i^r$ has linearly independent columns, so there exists a left inverse. In other words, we assume the mapping from action to feasible action (Defs. 2, 4) is one-to-one. This assumption is justified in the Appendix due to space limits.*

**Definition 2** (Feasible spaces of robots). *A feasible space, $H_i^r$, of robot i in round r is the set of feasible data-points the robot can send to the cloud:*

$$H_i^r = \{v_i^r = P_i^r a_i^r \mid \mathbf{1}^\top a_i^r \leq N_{cache}, a_i^r \in \mathbb{R}_+^{N_{class}}\}.$$

*$H_i^r$ is the convex hull of all columns of $P_i^r$ and $\mathbf{0}$. To simplify notation, $v_i^r = P_i^r a_i^r$ represents a feasible action $v_i^r$, which is obtained by multiplying an intended action $a_i^r$ by the feasible data matrix $P_i^r$.*

Intuitively, the feasible space (see Fig. 2) represents the expected number of datapoints uploaded per class but not the exact number due to perceptual uncertainty. Each robot uploads $N_{cache}$ data-points sampled from action $a_i^r$, which is pooled in the cloud. We assume we only get ground-truth labels $y$ in the cloud, since the limited cache of images can be scalably annotated by a human. Then, we re-train a new perception model on the new dataset $\mathscr{D}_c^{r+1}$. Each robot then downloads the new model parameters $\theta_i^{r+1}$, along with the new confusion matrix and latest cloud dataset distribution, $\rho_{\mathscr{D}_c^{r+1}}$. Our formulation is general – models and confusion matrices do not have to be updated every round $r$ and we can, for example, simply re-train a model after $M$ rounds of data collection.

**Collective Goal: Achieving a Target Data Distribution**   Often, we want to achieve a balanced dataset in the cloud with ample representation of rare events in order to train a robust ML model. As such, the shared goal of all the robots is to achieve *any* user-specified target dataset $\rho_{\mathscr{D}_{target}}$, which defines the number of data-points of each class the robots want to collect in the cloud. The fleet's goal is to choose actions $a_i^r$; $\forall i = 1, ..., N_{robot}$ at round $r$ to *collectively* reduce a strictly convex distance metric, denoted by $\mathscr{L}(\rho_{\mathscr{D}_c^r}, \rho_{\mathscr{D}_{target}})$, penalizing the difference between the current cloud dataset $\rho_{\mathscr{D}_c^r}$ and the target dataset $\rho_{\mathscr{D}_{target}}$. Our general framework can handle any strictly convex distance metric, such as the $L_2$ norm or the Kullback-Leibler (KL) Divergence [37]. Since all robots have a common goal to maximize the negative loss $-\mathscr{L}(\rho_{\mathscr{D}_c^r}, \rho_{\mathscr{D}_{target}})$, which is a concave potential function, our setting is a potential game with concave rewards (see Sec. 5.1).

**Centralized Oracle Action Policy:**   We now provide a formal optimization problem for distributed data collection. To provide key insight, we first describe a centralized "oracle" solution that has perfect information about all robots $i$, namely their confusion matrix $C_i^r$ and statistics of their data distribution $p_i^r(y)$. Then, we formalize a greedy, individualized approach and our interactive game-theoretic approach that matches the oracle policy's performance.

An oracle action policy, denoted by ORACLE, has access to all robots' data distributions and confusion matrices $C_i^r$. The oracle calculates each robot's action $a_i^r$ by solving the convex optimization problem in Eq. 1. The constraint (Eq. 1c) ensures that the actions $a_i^r$ do not exceed the cache limit $N_{cache}$. Eq. 1d shows the update of the cloud dataset for round $r+1$ based on the actions $a_i^r$ taken in the feasible space, $P_i^r a_i^r$, by each robot for round $r$, which we now detail.

A key subtlety is to update the cloud dataset $\rho_{\mathscr{D}_c^{r+1}}$ by merging the current cloud dataset $\rho_{\mathscr{D}_c^r}$ and each robots' uploaded dataset $a_i^r$. However, each robot's action is imperfect – it might think it is uploading class $j$ but due to perceptual uncertainty it might actually upload another class $j'$. Specifically, the robot's transmitted dataset $a_i^r$ is calculated from the predicted class labels $\hat{y}_j$ and not the true class labels $y_j$, which are not available on-robot. However, we can use predicted class probabilities $p_i^r(\hat{y}_j)$ to estimate true class probabilities $p_i^r(y_j)$ by: $p_i^r(y_j) = \sum_{k=1}^{N_{class}} p_i^r(\hat{y}_k) \cdot p_i^r(y_j|\hat{y}_k)$. Note that each robot only receives a confusion matrix $C_i^r$ from the cloud which consists of conditional probabilities $p_i^r(\hat{y}_j|y_j)$ and not $p_i^r(y_j|\hat{y}_j)$. Therefore, we still need to figure out a way to calculate $p_i^r(y_j|\hat{y}_j)$. Due to space limits, we present the Bayesian update of $p_i^r(y_j|\hat{y}_j)$ in the Appendix 5.3.

| PROBLEM 1: ORACLE OPTIMIZATION | PROBLEM 2: GREEDY OPTIMIZATION |
|---|---|

$$\min_{a_1^r \dots a_{N_{\text{robot}}}^r} \mathscr{L}(\rho_{\mathscr{D}_c^{r+1}}, \rho_{\mathscr{D}_{\text{target}}}) \tag{1a}$$

$$\text{subject to: } a_i^r \geq 0; \ \forall \ i = 1, \dots, N_{\text{robot}} \tag{1b}$$

$$1^T \cdot a_i^r \leq N_{\text{cache}}; \ \forall \ i = 1, \dots, N_{\text{robot}} \tag{1c}$$

$$\rho_{\mathscr{D}_c^{r+1}} = \rho_{\mathscr{D}_c^r} + \sum_{i=1}^{N_{\text{robot}}} (P_i^r a_i^r) \tag{1d}$$

$$\min_{a_i^r} \mathscr{L}(\rho_{\mathscr{D}_c^{r+1}}, \rho_{\mathscr{D}_{\text{target}}}) \tag{2a}$$

$$\text{subject to: } a_i^r \geq 0 \tag{2b}$$

$$1^T \cdot a_i^r \leq N_{\text{cache}} \tag{2c}$$

$$\rho_{\mathscr{D}_c^{r+1}} = \rho_{\mathscr{D}_c^r} + (P_i^r a_i^r) \tag{2d}$$

**Greedy Action Policy:** A greedy action policy, referred to as GREEDY, will not have any information about other robots' local data distribution, confusion matrix, or observed datasets. Thus, the best the robot can do is to attempt to minimize the loss function $\mathscr{L}(\rho_{\mathscr{D}_c^{r+1}}, \rho_{\mathscr{D}_{\text{target}}})$ by only optimizing its own action $a_i^r$ *individually*, as shown in Eq. 2. The optimization program 2 is very similar to that of the ORACLE policy (Eq. 1), with the only difference being that the decision variables are reduced to one. Since the ORACLE (Eq. 1) and GREEDY (Eq. 2) policy optimization programs have a convex objective with linear constraints, they are guaranteed to converge to an optimal solution.

## 3   A Cooperative Algorithm for Data Collection

We propose an INTERACTIVE algorithm for generating actions for each robot, which only requires interaction between the robots and no cloud coordination. Rather than the cloud calculating actions for each robot in one-shot, as shown in the ORACLE optimization program (1), each robot calculates its actions individually using shared information from other robots. Importantly, each robot only shares its feasible action without divulging its confusion matrix or local data distribution to others.

Alg. 1 describes our INTERACTIVE policy, which runs for each round $r$. The inputs (line 1), which are visible to each robot, are the target dataset $\rho_{\mathscr{D}_{\text{target}}}$ and the current cloud dataset $\rho_{\mathscr{D}_c^r}$. We initialize each robot's action $a_i^r$ in lines 2 - 4 using the GREEDY policy (Eq. 2) because the robots have not yet communicated any information about each others' tentative actions. In lines 5 - 10, we calculate optimal actions for each robot using the INTERACTIVE message passing algorithm.

We start by sharing each robot's product of feasible data matrix and initial action (line 3) with all other robots (line 5). Then, we iterate over each robot (lines 7 - 10) and calculate its best action $a_i^r$ using the optimization program Eq. 3 while considering the other robots' actions fixed (line 8). The optimization program in Eq. 3 is similar to that of the ORACLE policy (Eq. 1); the difference lies in the calculation of the cloud dataset at round $r + 1$ in Eq. 3d and having one decision variable.

In line 9, each robot shares its product of the feasible data matrix and the optimal action calculated using Eq. 3 with the others. This repeats until our system reaches a Nash equilibrium (i.e. a fixed point, where no robot would change its action). Finally, after convergence, we upload data from each robot sampled according to its final calculated action $a_i^r$ (line 13). Since the INTERACTIVE optimization program 3 is convex, it converges to an optimal solution (see Thm. 1).

1 **Input:** Target, Cloud Dataset $\rho_{\mathscr{D}_{\text{target}}}, \rho_{\mathscr{D}_c^r}$
2 **for** $i = 1, \dots, N_{\text{robot}}$ **do**
3     Initialize $a_i^r$ using GREEDY actions Eq. 2.
4 **end**
5 Share $P_i^r a_i^r$ with all robots.
6 **while** Not Converged **do**
7     **for** $i = 1, \dots, N_{\text{robot}}$ **do**
8        Get action $a_i^r$ using opt. program Eq. 3
9        Share actions $P_i^r a_i^r$ with all robots.
10     **end**
11 **end**
12 **for** $i = 1, \dots, N_{\text{robot}}$ **do**
13     Upload caches determined by actions $a_i^r$
14 **end**

**Algorithm 1: INTERACTIVE Algorithm**

| PROBLEM 3: INTERACTIVE OPTIMIZATION |
|---|

$$\min_{a_i^r} \mathscr{L}(\rho_{\mathscr{D}_c^{r+1}}, \rho_{\mathscr{D}_{\text{target}}}) \tag{3a}$$

$$\text{subject to: } a_i^r \geq 0 \tag{3b}$$

$$1^T \cdot a_i^r \leq N_{\text{cache}} \tag{3c}$$

$$\rho_{\mathscr{D}_c^{r+1}} = \rho_{\mathscr{D}_c^r} + \sum_{k=1; k \neq i}^{N_{\text{robot}}} (P_k^r a_k^r) + (P_i^r a_i^r) \tag{3d}$$

**Theoretical Analysis:** We first show that the *while* loop (lines 6 - 11) in our proposed Alg. 1 will eventually converge. Moreover, we provide easily-obtained conditions for when it converges in *one iteration*, which minimizes inter-robot communication. Crucially, we also show that our interactive policy matches the omniscient oracle policy. **All proofs** are in the Appendix 5.6 - 5.9.

**Theorem 1** (Convergence). *The while loop (lines 6 - 11) in Alg. 1 will eventually converge.*

Next, we show one of the main technical contributions of this paper, which states that our proposed INTERACTIVE algorithm will reach the same optimal solution as the ORACLE upon termination.

**Theorem 2** (INTERACTIVE converges to ORACLE). *The while loop in Alg. 1 lines 6 - 11 is guaranteed to return an action (denoted by $a_{int,i}^r$) that is equal to the ORACLE action denoted by $a_{o,i}^r$.*

Next, we provide practical conditions for when our proposed INTERACTIVE action policy will converge in *one iteration* of message passing, which bounds inter-robot communication.

**Theorem 3** (Bounded Communication). *When the total number of uploaded data-points is smaller than the difference between the size of target dataset $\mathscr{D}_{target}$ and the current cloud dataset $\mathscr{D}_c^r$, namely $\mathbf{1}^\top(\mathscr{D}_{target} - \mathscr{D}_c^r) > N_{robot} \times N_{cache}$, the while loop in Alg. 1 lines 6 - 11 terminates in one iteration.*

The condition in Thm. 3 holds for all rounds except for the last round that reaches the target distribution, upon which data collection terminates. All our theory assumes that all actions $a_i^r \in \mathbb{R}^{N_{class}}$ can realize any feasible real-valued vector. However, in reality, we will only have an integer-valued action vector since we can only upload a discrete set of images, which becomes an integer programming problem. However, for real-world datasets with thousands of images, we can just round the continuous solution to get a very close approximation to the (generally intractable) integer case.

## 4  Experiments and Conclusion

We now compare our Alg. 1 with benchmark methods on four diverse datasets. The first two datasets of `MNIST` [38] and `CIFAR-10` [39] serve as proof-of-concepts for the domains of handwritten digit and common object classification. Then, we use the `Adverse-Weather` dataset [40], which contains tens of thousands of images to train self-driving vehicles to classify rain, fog, snow, sleet, overcast, sunny, and cloudy driving conditions. To show the generality of our theory, we then extend to the state-of-the-art Berkeley Deep Drive (`DeepDrive`) dataset [6], which has 100K images of various weather conditions and road scenarios for self-driving cars.

*Comparison Metric:* To compare all methods, we use the $L_2$-norm (the optimization objective) between the target $\rho_{\mathscr{D}_{target}}$ and the current cloud dataset $\rho_{\mathscr{D}_c^r}$. For statistical confidence, all experiments are repeated for more than 10 times with different random seeds that capture uncertainty in sampling from the confusion matrix $C_i^r$ and observing different distributions of local data per robot. Further experiment parameters are detailed in the Appendix 5.4. We compare the following methods:

1. GREEDY solves the optimization program in Eq. 2 *individually* per robot by minimizing the $L_2$-norm between the target and cloud distribution without information about other robots.
2. ORACLE solves the optimization program in Eq. 1. It perfectly knows all incoming class data distributions $p_i^r(y)$ and confusion matrices $C_i^r$ for all robots $i$ and thus calculates the optimal action for each robot in one common optimization problem (Eq. 1).
3. UNIFORM is a deterministic policy which assigns the same probabilities to all classes for each robot, i.e. $a_i^r = \left[ \frac{1}{N_{class}} \cdots \frac{1}{N_{class}} \right]$. It represents a simple heuristic for equally sampling all classes.
4. LOWER-BOUND (derived in Lemma 6) is the lower bound of the objective function of the ORACLE policy for a given target dataset, $\rho_{\mathscr{D}_{target}}$, current cloud dataset, $\rho_{\mathscr{D}_c^r}$, and local data distribution $p_i^r(y)$. It represents how well can sample in the absence of perceptual uncertainty.
5. INTERACTIVE runs our Alg. 1. It is not an *Oracle* policy, as it only shares the action taken by other robots and not the actual class data distribution, $p_i^r(y)$, nor the confusion matrix.

**Results:** Our experimental results (Fig. 3) demonstrate that our proposed INTERACTIVE policy performs as well as the ORACLE , as proved in Thm. 2. Additionally, we demonstrate that our INTERACTIVE policy is much better than the GREEDY and UNIFORM policies on all datasets. Finally, we show that no action policy can perform better than the derived LOWER-BOUND action policy.

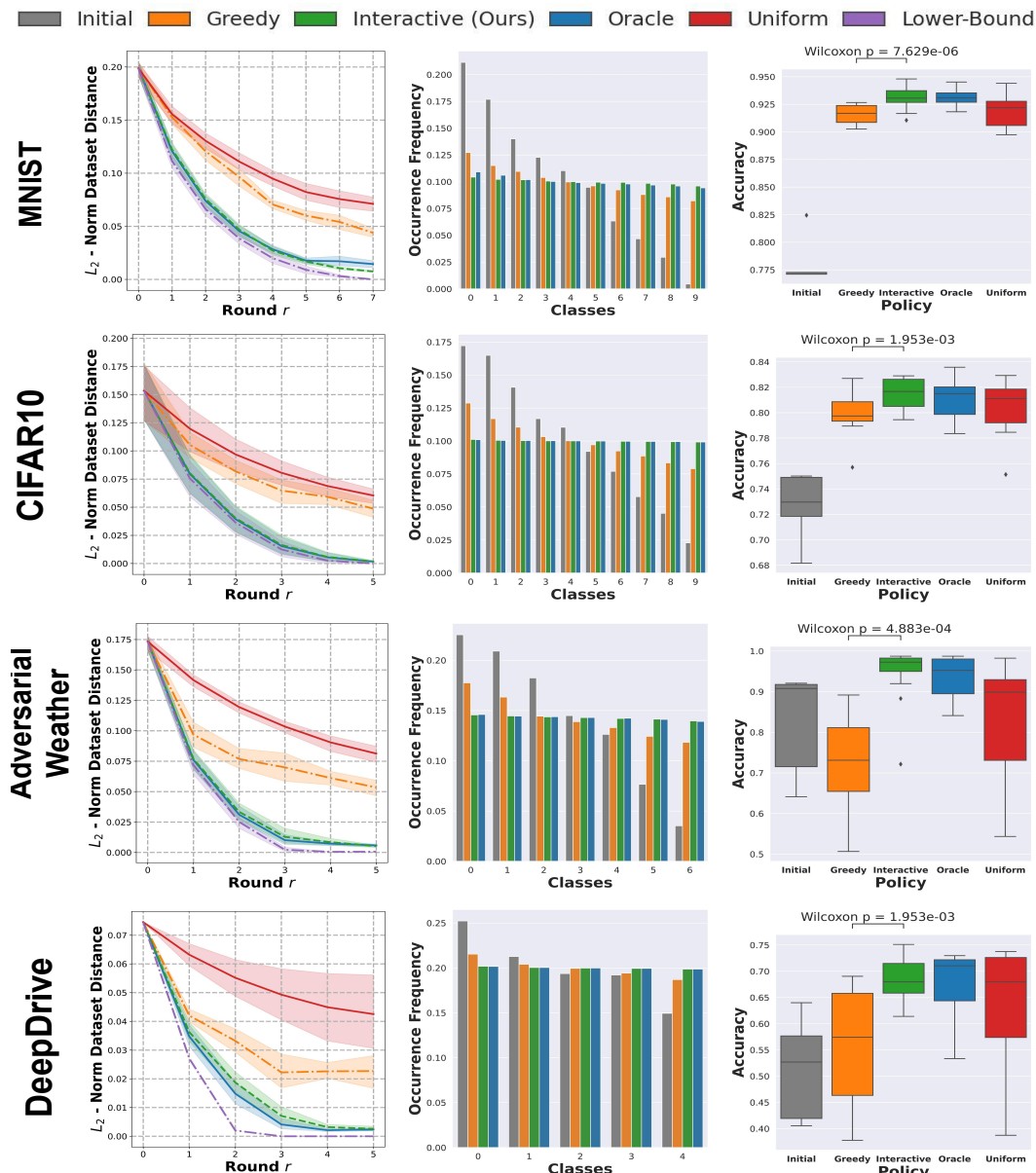

Figure 3: **Our game-theoretic INTERACTIVE policy outperforms benchmarks and converges to the OR-ACLE.** Each row is a different dataset. *Column 1:* As expected by our theory, INTERACTIVE minimizes the $L_2$-norm distance (optimization objective, y-axis) better than GREEDY and matches the omniscient ORACLE. *Column 2:* Clearly, INTERACTIVE achieves a much more balanced distribution of classes (target distribution is uniform) than benchmarks. *Column 3:* Since INTERACTIVE achieves a more balanced dataset, this experimentally translates to a higher DNN accuracy (statistically significant) on a held-out test dataset.

### *Does cooperation minimize distance to the target data distribution?*

Our optimization objective is to minimize the $L_2$-norm distance between the cloud dataset and target data distribution, which we plot in the first column. Clearly, our INTERACTIVE policy significantly outperforms the GREEDY and UNIFORM policies on all datasets. Specifically, we beat the GREEDY policy by $23.6\%, 44.8\%, 40.3\%, 38.7\%$ in $L_2$-norm distance on the MNIST, CIFAR-10, Adverse-Weather, and DeepDrive datasets respectively. These benefits arise because the INTER-ACTIVE policy allows robots to coordinate the rare classes they upload, but the GREEDY policy might lead to uncoordinated uploading of redundant data. Moreover, our INTERACTIVE policy performs nearly identically as the ORACLE method, with small deviations due to imperfect vision models and randomness in local data distributions between trials. This is natural since we proved

that the INTERACTIVE action policy reaches the same optimal value in expectation as the ORACLE policy in Thm. 2. Finally, we observe that no action policy outperforms our LOWER-BOUND policy derived in Lemma 6.

***Does cooperation achieve more balanced datasets?***

In column two, we see that the initial data distribution among robots (gray) is highly imbalanced since they operate in diverse contexts. However, we see that our INTERACTIVE policy (green) achieves a much more balanced dataset distribution compared to GREEDY (orange), which is natural since the convex objective minimizes the distance to a uniform distribution.

***What is the final accuracy of trained models?***

In column 3, we show the final accuracy of re-training DNN classification models on the datasets accrued by each method in the cloud. Importantly, our proposed INTERACTIVE action policy leads to better accuracy gains than the GREEDY and UNIFORM action policies. We beat the GREEDY policy by $1.4\%, 1.7\%, 21.9\%, 12.4\%$ in accuracy on the MNIST, CIFAR-10, Adverse-Weather, and DeepDrive datasets respectively. This is because the INTERACTIVE action policy makes sure we collect classes lacking in the current cloud dataset, thus preventing class-imbalance issues in model training. While our theory only addresses convex distances between dataset distributions (column 1 and 2), we show strong experimental results for re-training non-convex DNN classifiers. The INTERACTIVE and ORACLE algorithms lead to slightly different final accuracies since they can potentially upload a different set of images and there is not a closed form relationship between the number of images and accuracy of a non-convex DNN. As detailed in the Appendix, INTERACTIVE achieves very close to state-of-the-art accuracy for each dataset with only a limited set of uploaded datapoints. DNN architectures are also detailed in the Appendix. Collectively, these results closely align with our theory and show strong experimental benefits on real-world data.

**Limitations:** Our work assumes each robot can interact, which does not scale for extremely large fleets. Moreover, we assume that we sample images according to a classification model, even though we can train models for other tasks on the uploaded images. In future work, we aim to extend our theoretical guarantees for sub-clusters of communicating robots and cluster a continuous data distribution based on similar embeddings that serve as virtual "classes". Such an ability to generalize beyond discrete classes may enable our algorithm to scale to learning data-driven control policies.

**Conclusion:** This paper presents a theoretically-grounded, cooperative data sampling policy for networked robotic fleets, which converges to an oracle policy upon termination. Additionally, it converges in a single iteration under a mild practical assumption, which allows communication efficiency on real-world AV datasets. Our approach is a first step towards an increasingly timely problem as today's AV fleets measure terabytes of heterogenous data in diverse operating contexts [1]. In future work, we plan to develop policies that approximate the oracle solution when only a subset of robots can form coalitions and certify their resilience to adversarial node failures.

# Acknowledgements

This material is based upon work supported in part by Cisco Systems, Inc. under MRA MAG00000005. This material is also based upon work supported by the National Science Foundation under grant no. 2148186 and is supported in part by funds from federal agency and industry partners as specified in the Resilient & Intelligent NextG Systems (RINGS) program.

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
