# OpenReview forum: "Decentralized Data Collection for Robotic Fleet Learning: A Game-Theoretic Approach"
_robot-learning.org/CoRL/2022/Conference — CoRL 2022 Poster_

### Official Review · Reviewer_YAM6 · 2022-07-23

**Originality:** Good
**Technical Quality:** Very Good
**Clarity Of Presentation:** Good
**Impact:** 3

**Recommendation:**

Weak Accept: I recommend accepting the paper, but will not argue for my recommendation if the majority of other reviewers have a different opinion.

**Summary:**

This paper studies distributed data collection by autonomous robots, in which a large number of robots aim to collaboratively collect the data such that the joint dataset matches some target distribution. The proposed algorithm cast this problem as a potential game, and hence propose an algorithm which does not require centralised access to all information yet achieves comparable performance.

**Issues:**

Please see "Strengths And Weaknesses" for details.

**Quality Of The Limitations Section:**

Limitations are addressed clearly

**Reviewer Expertise:**

2: The reviewer is willing to defend the evaluation, but it is quite likely that the reviewer did not understand central parts of the paper

**Robotics Focus:**

Highly relevant to robotics but no hardware experiments

**Strengths And Weaknesses:**

Strengths:
- The approach of casting the problem of decentralized data collection as a game is interesting.
- The approach taken for the proposed algorithm, i.e., firstly giving a centralised algorithm with perfect information and then relaxing it to derive a practical algorithm with less required information, is reasonable.
- The empirical results are promising, and indeed shows that the performance of the proposed method can match that of the ideal algorithm with oracle access.

Weaknesses:
- I'm not entirely sure about the practical benefit of the proposed algorithm compared with the oracle algorithm with perfect information. If I understand correctly, the oracle algorithm can be achieved by letting every agent share their information with the central coordinator, and the central coordinator can then do all the calculation; the proposed INTERACTIVE algorithm does not require sending all information to the central coordinator yet it still requires the robots to share information with each other. I wonder is it possible to give some justification as to whether the proposed algorithm requires less computation than the oracle algorithm?

**Summary Of Recommendation:**

The paper takes a nicely grounded approach to solve an interesting problem from a game-theoretical perspective.

---

### Official Review · Reviewer_scJ6 · 2022-07-30

**Originality:** Good
**Technical Quality:** Very Good
**Clarity Of Presentation:** Good
**Impact:** 3

**Recommendation:**

Weak Accept: I recommend accepting the paper, but will not argue for my recommendation if the majority of other reviewers have a different opinion.

**Summary:**

This paper poses the following question: in a setting with a large number of autonomous vehicles that operate under communication bandwidth constraints, what data should each vehicle send to a central server (e.g., for labeling and training a machine learning model to be used by the vehicles for tasks such as image classification)? If there were no bandwidth constraints and no constraints on labeling costs on the central server, each vehicle would send all its data. In a practical setting this is not possible. The challenge is to have the vehicles send a diverse set of data such that central server sees as 'balanced' a data distribution as possible. The paper casts this distributed data collection problem as a potential game and shows that the (distributed) strategy converges to a centralized 'oracle' policy. The paper also gives theoretical performance bounds and compares with other (albeit simpler) strategies for the problem on an autonomous driving dataset.


**Issues:**

1. I would encourage the authors to replace swarms with fleets (or some other word). To often in robotics, people simply say swarm when they mean a large number. A swarm is more than just a large number of things. The morning rush hour traffic in a major city has a very large number of vehicles, but they do not constitute a swarm in the way that a large number of bees migrating to a new location or a large number of tourists suddenly descending upon the Louvre.

2. Is there a contradiction between convex objective function (line 37) and concave objective function (line 59)?

3. What happens if the number of classes is unknown (section 2) ?



**Quality Of The Limitations Section:**

Limitations are addressed clearly

**Reviewer Expertise:**

4: The reviewer is confident but not absolutely certain that the evaluation is correct

**Robotics Focus:**

Highly relevant to robotics but no hardware experiments

**Strengths And Weaknesses:**

The paper has the following strengths:
1. The formulation is novel (at least to this reviewers knowledge) in this setting (autonomous vehicles uploading data to the cloud over a bandwidth constrained channel).
2. The proof of convergence of the INTERACTIVE algorithm (and the convergence to ORACLE).
3. Experimentally, INTERACTIVE achieves a more balanced distribution of classes.

Food for thought:
1. The paper says that the algorithm converges under a mild practical assumption. This is true in the setting explored by the authors, but the real problem is that the setting itself is vastly oversimplified due to the assumption that all vehicles may communicate with each other. The authors acknowledge this in the limitation section.
2. I would have liked to see a discussion of a more general set of tasks than image classification. What if the goal of data collection was to improve the control policy on each vehicle instead of simply having better image classification tomorrow than was available today?
3. It is not clear to me that an agreed upon fixed round at which updates are sent is needed. Would things change (possibly improve) if some vehicles decided to send updates at lower frequencies?

**Summary Of Recommendation:**

I think this is a nice piece of work with interesting ideas. It is well situated in the body of work that asks questions about how autonomous vehicles should use the cloud efficiently (i.e. while respecting constraints e.g., on bandwidth and labeling costs). If one takes a broad view of 'robotics' (which I do), then this work is certainly relevant to the field of robotics. I am not enough of an expert to know whether the ideas in the paper have been developed in other contexts and are simply novel to robotics people or are more fundamentally new which is really the only reason I marked the paper as a weak accept instead of a strong accept.

---

### Official Review · Reviewer_hwcw · 2022-08-02

**Originality:** Fair
**Technical Quality:** Good
**Clarity Of Presentation:** Good
**Impact:** 2

**Recommendation:**

Strong Reject: I recommend rejecting the paper and will argue for my recommendation even if other reviewers hold a different opinion.

**Summary:**

The paper is written in the context of multiple-agents that collect data in the field. They consider the problem where the data collected by these agents need to be aggregated in a central location in order to train models on the data. As bandwidth might be limited the agents cannot transfer all the data. Hence the authors propose an algorithm that sub-samples the data to only send a fraction of the entire dataset over the network. They claim their proposed methods outperforms baselines while maintaining a target distribution required at the central location.

**Issues:**

NA

**Quality Of The Limitations Section:**

Limitations are addressed clearly

**Reviewer Expertise:**

3: The reviewer is fairly confident that the evaluation is correct

**Robotics Focus:**

Highly relevant to robotics but no hardware experiments

**Strengths And Weaknesses:**

Strengths
1) The proposed approach is quite general and has some convergence guarantees theoretically.

Weaknesses
1) The experimental section of the paper is not quite related with robotics. The datasets collected by multiple robots in the field will likely not have IID distribution. The datasets considered in this evlatuation section of this work have IID distribution. Any proposed method to tackle this problem should consider experiments where the distributions of each agent might be quite different from each other but have strong correlation between the datapoints on one robot. For example, robot 1 would see picture of rain for quite some time while robot 2 might be in an area where it is sunny.

2) How is the model initialized in the data collection process? It seems the agent needs access to a model with non-trivial accuracy in order to choose data samples correctly.

3) How does accuracy change if there is a drift because of errors made by an inaccurate model?

**Summary Of Recommendation:**

My recommendation is mostly due to the experimental section not considering scenarios more relevant to robotics setup.

---

### Official Review · Reviewer_2cKZ · 2022-08-03

**Originality:** Very Good
**Technical Quality:** Very Good
**Clarity Of Presentation:** Very Good
**Impact:** 3

**Recommendation:**

Weak Accept: I recommend accepting the paper, but will not argue for my recommendation if the majority of other reviewers have a different opinion.

**Summary:**

This paper presents cooperative data sampling policy for networked robotic fleets, which converges to an oracle policy upon termination. The motivation is the fact that a lot of AV problems would collect a lot of data, and we cannot quite upload all the data to a common database, so we would want to somehow learn from all the data without actually transferring it to the cloud. The authors cast this problem as a co-operative game, and under some linearity conditions derive an approach that performs well.

**Issues:**

See weaknesses.

**Quality Of The Limitations Section:**

Limitations section not present

**Reviewer Expertise:**

2: The reviewer is willing to defend the evaluation, but it is quite likely that the reviewer did not understand central parts of the paper

**Robotics Focus:**

Highly relevant to robotics but no hardware experiments

**Strengths And Weaknesses:**

Strengths:

1. The problem being studied is interesting and real, especially at scale from an AV company point of view.

2. I liked the experiments in the paper -- they start from simple, but intuitive CIFAR and MNIST experiments, but scale it up to BDD-level experiments, which is interesting It would be great to see this scaled up even further, since this strategy really matters when scaling.

Weaknesses:

1. Limitations section is not present, which is compulsory, though I hope that authors will revise the paper to add it.

Unfortunately, I am not an expert in this area, and have only very limited knowledge of the related works. The idea of framing into a co-operative game is interesting, but I don't know if that's studied already or even in general, how is distributed data collection posed.

**Summary Of Recommendation:**

I am unfortunately not very knowledgeable in this area, and would have to rely on other reviewers to assess the novelty and systems impact of this work. That said, I did read through the paper and checked the experiments, and briefly the theorems, and they do make sense to me. That's why I am going with a weak accept, with a confidence of 2.

---

### Meta-Review · Area_Chair_C9qf · 2022-08-11

**Recommendation:** Accept (Poster)
**Confidence:** 3

**Metareview:**

The paper is of high quality and clearly written. The game theoretic approach with theoretical proofs is interesting and empirical results support the theory. Reviewers (3 x weak accept) are leaning towards accepting the paper. While reviewer hwcw's original score is strong reject, based on the discussions with the authors and the authors' latest changes it seems the desire to reject is not as strong anymore. Therefore, I recommend acceptance (Poster).

**Best Paper Nomination:**

No